# Fast Computation of Leave-One-Out Cross-Validation for $k$-NN Regression

**Motonobu Kanagawa**  *motonobu.kanagawa@eurecom.fr*
*Data Science Department*
*EURECOM*

**Reviewed on OpenReview:** *https://openreview.net/forum?id=SBE2q9qwZj*

## Abstract

We describe a fast computation method for leave-one-out cross-validation (LOOCV) for $k$-nearest neighbours ($k$-NN) regression. We show that, under a tie-breaking condition for nearest neighbours, the LOOCV estimate of the mean square error for $k$-NN regression is identical to the mean square error of $(k+1)$-NN regression evaluated on the training data, multiplied by the scaling factor $(k+1)^2/k^2$. Therefore, to compute the LOOCV score, one only needs to fit $(k+1)$-NN regression only once, and does not need to repeat training-validation of $k$-NN regression for the number of training data. Numerical experiments confirm the validity of the fast computation method.

## 1 Introduction

$k$-Nearest Neighbours ($k$-NN) regression (Stone, 1977) is a classic nonparametric regression method that often performs surprisingly well in practice despite its simplicity (Chen et al., 2018) and thus has been actively studied both theoretically and methdologically (e.g, Györfi et al., 2002; Kpotufe, 2011; Jiang, 2019; Azadkia, 2019; Madrid Padilla et al., 2020; Kpotufe and Martinet, 2021; Lalande and Doya, 2023; Ignatiadis et al., 2023; Matabuena et al., 2024). It has a wide range of applications such as missing-value imputation (Troyanskaya et al., 2001), conditional independence testing (Runge, 2018), outlier detection (Breunig et al., 2000), approximate Bayesian computation (Biau et al., 2015), and function-valued regression (Lian, 2011), to just name a few.

For any test input, $k$-NN regression obtains its $k$-nearest training inputs and averages the corresponding $k$ training outputs to predict the test output. Thus, the number $k$ of nearest neighbours (and the distance function on the input space) is a key hyperparameter of $k$-NN regression and must be selected carefully. Indeed, theoretically, it is known that $k$ should increase as the training data size $n$ increases for $k$-NN regression to converge to the true function (e.g., Györfi et al., 2002, Theorem 6.2). Hence, one should not use a prespecified value for $k$ (e.g., $k = 5$) and must select $k$ depending on the training data.

A standard way for selecting the hyperparameters of a learning method is cross-validation (e.g., Hastie et al., 2009, Section 7.10). However, cross-validation can be costly when the training data size is large. In particular, leave-one-out cross-validation (LOOCV) (Stone, 1974) can be computationally intensive since, if naively applied, it requires training the learning method on a training dataset of size $n-1$ and repeating it $n$ times. This issue also applies to the use of LOOCV for $k$-NN regression. On the other hand, theoretically, LOOCV is known to be better than cross-validation of a smaller number of split-folds as an estimator of the generalization error (e.g., Arlot and Celisse, 2010, Section 6.2). Azadkia (2019) theoretically analyses LOOCV for $k$-NN regression in selecting $k$ and discusses its optimality.

This paper describes a fast method for computing LOOCV for $k$-NN regression. Specifically, we show that, under a tie-breaking assumption for nearest neighbours, the LOOCV estimate of the mean-squared error is identical to *the mean square error of $(k+1)$-NN regression evaluated on the training data, multiplied by the scaling factor $(k+1)^2/k^2$* (Corollary 1 in Section 3). Therefore, to perform LOOCV, one only needs to fit

$(k + 1)$-NN regression *only once* and does not need to repeat the $k$-NN fit $n$ times. To our knowledge, this method has not been reported in the literature.

This paper is organised as follows. We describe $k$-NN regression and LOOCV in Section 2. We present the fast computation method in Section 3, empirically confirm its validity in Section 4, discuss the tie-breaking condition in Section 5, and conclude in Section 6.

## 2 $k$-NN Regression and LOOCV

Suppose we are given $n$ input-output pairs as training data:

$$D_n := \{(x_1, y_1), \ldots, (x_n, y_n)\} \subset \mathcal{X} \times \mathcal{Y}, \tag{1}$$

where $\mathcal{X}$ and $\mathcal{Y}$ are the input and output spaces. Specifically, $\mathcal{X}$ is a metric space with a distance metric $d_{\mathcal{X}} : \mathcal{X} \times \mathcal{X} \mapsto [0, \infty)$ (e.g., $\mathcal{X} = \mathbb{R}^D$ is a $D$-dimensional space and $d_{\mathcal{X}}(x, x') = \|x - x'\|$ is the Euclidean distance). The output space $\mathcal{Y}$ can be discrete (e.g., $\mathcal{Y} = \{0, 1\}$ in the case of binary classification) or continuous (e.g., $\mathcal{Y} = \mathbb{R}$ for the case of real-valued regression, or $\mathcal{Y} = \mathbb{R}^M$ for vector-valued regression with $M$ outputs). Below, we use the following notation for the set of training inputs:

$$X_n := \{x_1, \ldots, x_n\}. \tag{2}$$

Assuming that there is an unknown function $f : \mathcal{X} \to \mathcal{Y}$ such that

$$y_i = f(x_i) + \varepsilon_i, \quad i = 1, \ldots, n,$$

where $\varepsilon_1, \ldots, \varepsilon_n \in \mathcal{Y}$ are independent zero-mean noise random variables, the task is to estimate the function $f$ based on the training dataset $D_n$. $k$-NN regression is a simple, nonparametric method for this purpose, which often performs surprisingly well in practice and has solid theoretical foundations (e.g, Györfi et al., 2002; Kpotufe, 2011; Chen et al., 2018).

### 2.1 $k$-NN Regression

Let $k \in \mathbb{N}$ be fixed. For an arbitrary test input $x^* \in \mathcal{X}$, $k$-NN regression predicts its output by first searching for the $k$-nearest neighbours of $x^*$ from $X_n$ and then computing the average of the corresponding $k$ training outputs. To be more precise, define the set of indices for $k$ nearest neighbours as

$$\mathrm{NN}(x^*,\ k,\ X_n) := \{\ i_1, \ldots, i_k \in \{1, \ldots, n\}\ | \\ d_{\mathcal{X}}(x^*, x_{i_1}) \leq \cdots \leq d_{\mathcal{X}}(x^*, x_{i_k}) \leq d_{\mathcal{X}}(x^*, x_j) \text{ for all } j \in \{1, \ldots, n\} \backslash \{i_1, \ldots, i_k\}\ \}. \tag{3}$$

Then, the $k$-NN prediction for $x^*$ is defined as

$$\hat{f}_{k, D_n}(x^*) := \frac{1}{k} \sum_{i \in \mathrm{NN}(x^*,\ k,\ X_n)} y_i.$$

In the following, we make the following tie-breaking assumption, which makes $\mathrm{NN}(x_\ell,\ k,\ X_n)$ uniquely specified for $\ell = 1, \ldots, n$ (i.e., the inequalities '$\leq$' in (3) with $x^* = x_\ell$ become '$<$').

**Assumption 1** (Tie-breaking). *For all $i, j = 1, \ldots, n$ with $i \neq j$, we have $x_i \neq x_j$ and $d_{\mathcal{X}}(x_\ell, x_i) \neq d_{\mathcal{X}}(x_\ell, x_j)$ for all $\ell = 1, \ldots, n$.*

This tie-breaking condition is usually satisfied if $x_1, \ldots, x_n$ are multivariate and sampled from a continuous distribution. Section 5 provides a further discussion.

## 2.2 Leave-One-Out Cross-Validation (LOOCV)

LOOCV for $k$-NN regression is defined as follows. Consider the case $\mathcal{Y} = \mathbb{R}^M$ with $M \in \mathbb{N}$ ($M = 1$ is the case of standard real-valued regression). For each $\ell = 1, \ldots, n$, consider the training dataset (1) with the $\ell$-th pair $(x_\ell, y_\ell)$ removed:

$$D_n \backslash \{(x_\ell, y_\ell)\} = \{(x_1, y_1), \ldots, (x_{\ell-1}, y_{\ell-1}), (x_{\ell+1}, y_{\ell+1}), \ldots, (x_n, y_n)\}.$$

Then, the $k$-NN prediction for any $x^*$ based on $D_n \backslash \{(x_\ell, y_\ell)\}$ is

$$\hat{f}_{k, D_n \backslash \{(x_\ell, y_\ell)\}}(x^*) = \frac{1}{k} \sum_{i \in \mathrm{NN}(x^*, \ k, \ X_n \backslash \{x_\ell\})} y_i.$$

Then, the LOOCV score for $k$-NN regression can be defined as

$$\mathrm{LOOCV}(k, D_n) := \frac{1}{n} \sum_{\ell=1}^{n} \left\| \hat{f}_{k, D_n \backslash \{(x_\ell, y_\ell)\}}(x_\ell) - y_\ell \right\|^2. \tag{4}$$

That is, for each $\ell = 1, \ldots, n$, the held-out pair $(x_\ell, y_\ell)$ is used as validation data for the $k$-NN regression fitted on the training data $D_n \backslash \{(x_\ell, y_\ell)\}$ of size $n - 1$. Calculating the LOOCV score for a large $n$ is computationally expensive since one needs to fit $k$-NN regression $n$ times if naively implemented. Next, we will show how this computation can be done efficiently by fitting $(k + 1)$-NN regression only once.

## 3 Fast Computation of LOOCV for $k$-NN Regression

To compute the LOOCV score (4), we need to compute, for each $\ell = 1, \ldots, n$,

$$\hat{f}_{k, D_n \backslash \{(x_\ell, y_\ell)\}}(x_\ell) = \frac{1}{k} \sum_{i \in \mathrm{NN}(x_\ell, \ k, \ X_n \backslash \{x_\ell\})} y_i. \tag{5}$$

To this end, we need to obtain $\mathrm{NN}(x_\ell, \ k, \ X_n \backslash \{x_\ell\})$, the indices for the $k$ nearest neighbours of $x_\ell$ in $X_n \backslash \{x_\ell\}$.

The key insight is the following simple fact: *The union of $\{x_\ell\}$ and the $k$ nearest neighbours of $x_\ell$ in $X_n \backslash \{x_\ell\}$ is identical to the $k + 1$ nearest neighbours of $x_\ell$ in $X_n$.* That is, under Assumption 1, we have

$$\mathrm{NN}(x_\ell, \ k, \ X_n \backslash \{x_\ell\}) \cup \{x_\ell\} = \mathrm{NN}(x_\ell, \ k+1, \ X_n). \tag{6}$$

See Figure 1 for an illustration of the case $k = 3$. The reasoning is as follows. The first nearest neighbour of $x_\ell$ in $X_n$ is, of course, $x_\ell \in X_n$. The second nearest neighbour of $x_\ell$ in $X_n$ is the first nearest neighbour of $x_\ell$ in $X_n \backslash \{x_\ell\}$. Generally, for $1 \leq m \leq k$, the $(m + 1)$-th nearest neighbour of $x_\ell$ in $X_\ell$ is the $m$-th nearest neighbour of $x_\ell$ in $X_n \backslash \{x_\ell\}$. Therefore, the identity (6) holds.

Using (6), we can rewrite (5) in terms of $(k + 1)$-NN regression, as summarized as follows.

**Lemma 1.** *Under Assumption 1, we have, for all $k \in \mathbb{N}$ and $\ell = 1, \ldots, n$,*

$$\hat{f}_{k, D_n \backslash \{(x_\ell, y_\ell)\}}(x_\ell) = \frac{k+1}{k} \hat{f}_{k+1, D_n}(x_\ell) - \frac{1}{k} y_\ell$$

*Proof.* Using (6), we have

$$\begin{aligned}
\hat{f}_{k, D_n \backslash \{(x_\ell, y_\ell)\}}(x_\ell) &= \frac{1}{k} \sum_{i \in \mathrm{NN}(x_\ell, \ k, \ X_n \backslash \{x_\ell\})} y_i = \frac{1}{k} \left( \sum_{i \in \mathrm{NN}(x_\ell, \ k+1, \ X_n)} y_i - y_\ell \right) \\
&= \frac{k+1}{k} \frac{1}{k+1} \sum_{i \in \mathrm{NN}(x_\ell, \ k+1, \ X_n)} y_i - \frac{1}{k} y_\ell \\
&= \frac{k+1}{k} \hat{f}_{k+1, D_n}(x_\ell) - \frac{1}{k} y_\ell.
\end{aligned}$$

$\square$

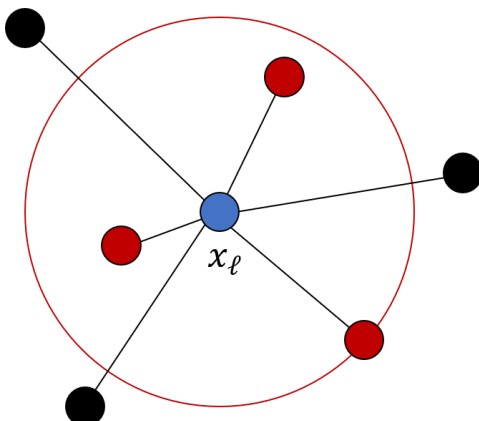

Figure 1: Illustration of $k = 3$ nearest neighbours of $x_\ell$. The blue point represents $x_\ell$, the three red points are the $k = 3$ nearest neighbours of $x_\ell$ in $X_n \backslash \{x_\ell\}$, the black points are other points in $X_n \backslash \{x_\ell\}$, and the red circle is the sphere of radius equal to the distance between $x_\ell$ and its third nearest neighbour in $X_n \backslash \{x_\ell\}$ (the red point on the circle).

Lemma 1 shows that the leave-one-out $k$-NN prediction $\hat{f}_{k,D_n \backslash \{(x_\ell,y_\ell)\}}(x_\ell)$ on the held-out input $x_\ell$ can be written as $\frac{k+1}{k} \hat{f}_{k+1,D_n}(x_\ell) - \frac{1}{k} y_\ell$, which does *not* involve the hold-out operation of removing $(x_\ell, y_\ell)$ from $D_n$; it just requires fitting the $(k+1)$-NN regression $\hat{f}_{k+1,D_n}$ on $D_n$, evaluate it on $x_\ell$, scale it by $(k+1)/k$ and subtract it by $y_\ell/k$.

Thus, to compute the LOOCV score (4), one needs to fit the $(k+1)$-NN regression *only once* on the dataset $D_n$. The resulting computationally efficient formula for the LOOCV score is given below.

**Corollary 1.** *Under Assumption 1, for the LOOCV score in* (4)*, we have*

$$\text{LOOCV}(k, D_n) = \left(\frac{k+1}{k}\right)^2 \frac{1}{n} \sum_{\ell=1}^{n} \left\| \hat{f}_{k+1,D_n}(x_\ell) - y_\ell \right\|^2. \tag{7}$$

*Proof.* Using Lemma 1, we have

$$\text{LOOCV}(k, D_n) = \frac{1}{n} \sum_{\ell=1}^{n} \left\| \hat{f}_{k,D_n \backslash \{(x_\ell,y_\ell)\}}(x_\ell) - y_\ell \right\|^2$$

$$= \frac{1}{n} \sum_{\ell=1}^{n} \left\| \frac{k+1}{k} \hat{f}_{k+1,D_n}(x_\ell) - \frac{1}{k} y_\ell - y_\ell \right\|^2 = \left(\frac{k+1}{k}\right)^2 \frac{1}{n} \sum_{\ell=1}^{n} \left\| \hat{f}_{k+1,D_n}(x_\ell) - y_\ell \right\|^2.$$

$\square$

The expression 7 shows that the LOOCV score (4) for $k$-NN regression is simply the *residual sum of squares* (or mean-square error) of the $(k+1)$-NN regression fitted and evaluated on $D_n$, multiplied by the scaling factor $\left(\frac{k+1}{k}\right)^2$. This scaling factor becomes large when $k$ is small, penalising overfitting and preventing too small $k$ from being chosen by LOOCV.

## 4  Experiments

We empirically check the validity of the formula (7) for efficient LOOCV computation.[1] We consider a real-valued regression problem where $\mathcal{X} = \mathbb{R}^d$ and $\mathcal{Y} = \mathbb{R}$, using two real datasets from `scikit-learn`: "Diabetes" and "Wine". We standardized each input feature to have mean zero and unit variance.

---

[1]The code for reproducing the experiments is available on `https://github.com/motonobuk/LOOCV-kNN`

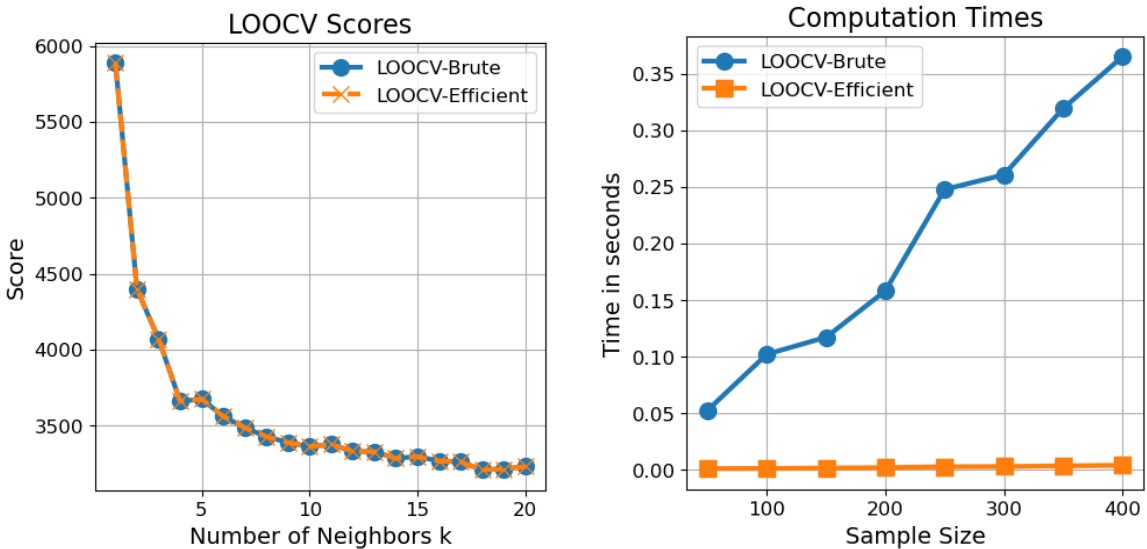

Figure 2: Experimental results on the Diabetes dataset. The left figure shows the LOOCV scores (4) computed in the brute-force manner ("LOOCV-Brute") and by using the derived formula (7) for different values of $k$. The right figure shows the computation times of either approach for different data sizes $n$ for fixed $k = 5$.

We compare two approaches: one is the brute-force computation of the LOOCV score (4) ("LOOCV-Brute"), and the other is the efficient computation based on the derived formula (7) ("LOOCV-Efficient"). We use the implementation of `scikit-learn`[2] with 'kd_tree' for computing nearest neighbours.

Figure 2 (Diabetes) and Figure 3 (Wine) show the respective results. On the left of each figure, we show the LOOCV scores computed by the two methods for different values of $k$, the number of nearest neighbours. They exactly coincide; we also have checked this numerically. This result verifies the correctness of the formula (7). On the right, we show the computation times[3] required for either method for different training data sizes $n$ for fixed $k = 5$. While LOOCV-Brute's computation time increases linearly with the sample size $n$, LOOCV-Efficient's computation time remains almost unchanged and is negligible compared with LOOCV-Brute's computation time. This shows the effectiveness of the formula (7) for the fast computation of the LOOCV score.

## 5 Discussion on the Tie-breaking Condition

Lastly, as a caveat, we consider the case where the tie-break condition in Assumption 1 is not satisfied. We use the same Diabetes and Wine datasets as in Section 4, but only employ one input feature in each dataset: "BMI" for the Diabetes and "malic-acid" for the Wine. Since each feature has many duplicates, there are many $i \neq j$ with $x_i = x_j$, thus violating the tie-breaking condition. Figure 4 shows the results corresponding to the left figures of Figures 2 and 3. LOOCV-Efficient is no longer exact and overestimates the LOOCV-Brute when $k$ is small.

This phenomenon can be understood by considering the case where $k = 1$. Suppose there is $\ell \in \{1, \ldots, n\}$ such that $x_\ell = x_i = x_j$ for some $i \neq j \neq \ell$. In this case, the first nearest neighbour of $x_\ell$ in $X_n = \{x_1, \ldots, x_n\}$ is $x_\ell$, $x_i$ and $x_j$, since $d_{\mathcal{X}}(x_\ell, x_\ell) = d_{\mathcal{X}}(x_\ell, x_i) = d_{\mathcal{X}}(x_\ell, x_j) = 0$. Therefore, depending on the tie-breaking rule of the nearest neighbour search algorithm, $x_i$ and $x_j$ may be selected as the first $k + 1 = 2$ nearest neighbours of $x_\ell$ in $X_n$; in this case the identity in (6) does not hold.

---

[2] https://scikit-learn.org/stable/modules/generated/sklearn.neighbours.KneighboursRegressor.html
[3] CPU: 1.1 GHz Quad-Core Intel Core i5. Memory: 8 GB 3733 MHz LPDDR4X.

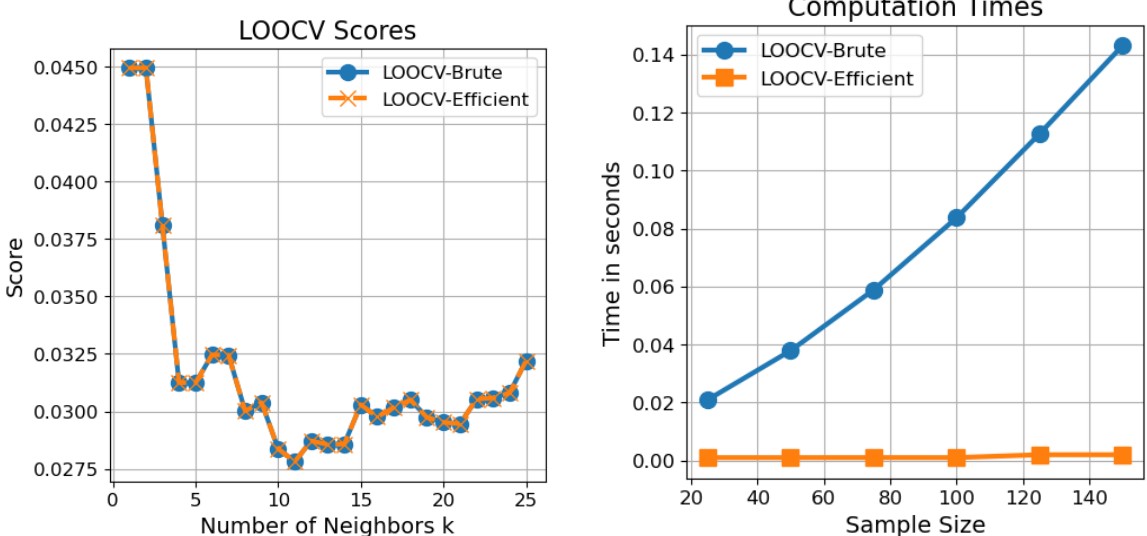

Figure 3: Experimental results on the Wine dataset. The left figure shows the LOOCV scores (4) computed in the brute-force manner ("LOOCV-Brute") and by using the derived formula (7) for different values of $k$. The right figure shows the computation times of either approach for different data sizes $n$ for fixed $k = 5$.

However, as shown in Figure 4, LOOCV-Efficient overestimates LOOCV-Brute only for *small $k$*, where the LOOCV scores are *large*. Therefore, this issue would not severely affect the *best $k$* that minimizes the LOOCV score. Indeed, for the Diabetes data, the best $k$ selected by LOOCV-Efficient is $k = 17$ and identical to LOOCV-Brute. For the Wine data, the best $k$ of LOOCV-Efficient is $k = 17$, while that of LOOCV-Brute is $k = 21$, which are similar and would not significantly change the prediction performance.

The above example suggests that Assumption 1 is essential for the formula (7) to be exact. In practice, one could check whether Assumption 1 is satisfied by, e.g., checking whether there are no duplicates in the inputs $x_1, \ldots, x_n$ beforehand; if they exist, one could resolve the duplicates by, e.g., taking the average of the outputs of duplicated samples. However, as demonstrated in the results in Figures 2 and 3 for the full Diabetes dataset, Assumption 1 would be satisfied for many practical situations where the input features are multivariate and continuous.

## 6  Conclusions

We showed that LOOCV for $k$-NN can be computed quickly by fitting $(k + 1)$-NN regression only once, evaluating the mean-square error on the training data and multiplying it by the scaling factor $(k + 1)^2/k^2$. By applying this technique, many applications of $k$-NN regression can be accelerated. It also opens up the possibility of using LOOCV for optimising not only $k$ but also the distance function $d_{\mathcal{X}}$; this would be one important future direction.

### Acknowledgements

The author thanks Takafumi Kajihara for a discussion, as well as the Action Editor and the anonymous reviewers for their time and comments.

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

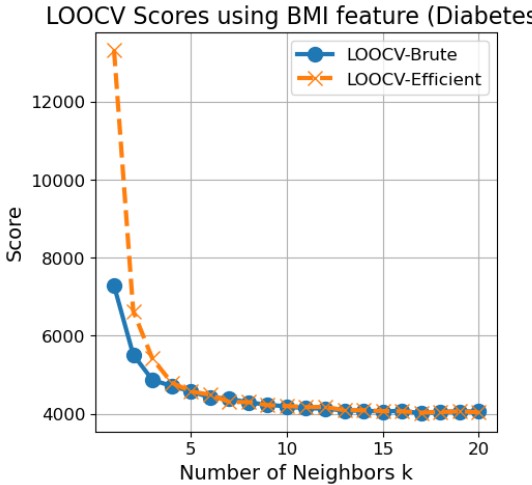
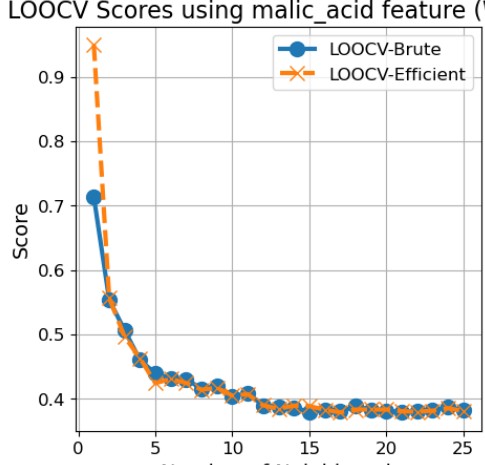

Figure 4: LOOCV scores for the Diabetes dataset (left) and the Wine dataset (right), each of which only uses *one input feature*. The used input feature has many duplicates and thus does not satisfy the tie-breaking condition in Assumption 1. **Left:** The best $k$ with the lowest LOOCV score is 17 for both LOOCV-Brute and LOOCV-Efficient. **Right:** The best $k$ with the lowest LOOCV score is 21 for LOOCV-Brute and 17 for LOOCV-Efficient.

Biau, G., Cérou, F., and Guyader, A. (2015). New insights into approximate Bayesian computation. In *Annales de l'IHP Probabilités et Statistiques*, volume 51, pages 376–403.

Breunig, M. M., Kriegel, H.-P., Ng, R. T., and Sander, J. (2000). LOF: identifying density-based local outliers. In *Proceedings of the 2000 ACM SIGMOD International Conference on Management of Data*, pages 93–104.

Chen, G. H., Shah, D., et al. (2018). Explaining the success of nearest neighbor methods in prediction. *Foundations and Trends® in Machine Learning*, 10(5-6):337–588.

Györfi, L., Kohler, M., Kryzak, A., and Walk, H. (2002). *A Distribution-Free Theory of Nonparametric Regression*. Springer.

Hastie, T., Tibshirani, R., and Friedman, J. (2009). *The Elements of Statistical Learning: Data Mining, Inference, and Prediction*. Springer Science & Business Media.

Ignatiadis, N., Saha, S., Sun, D. L., and Muralidharan, O. (2023). Empirical Bayes mean estimation with nonparametric errors via order statistic regression on replicated data. *Journal of the American Statistical Association*, 118(542):987–999.

Jiang, H. (2019). Non-asymptotic uniform rates of consistency for k-NN regression. In *Proceedings of the AAAI Conference on Artificial Intelligence*, volume 33, pages 3999–4006.

Kpotufe, S. (2011). k-NN regression adapts to local intrinsic dimension. *Advances in Neural Information Processing Systems*, 24.

Kpotufe, S. and Martinet, G. (2021). Marginal singularity and the benefits of labels in covariate-shift. *The Annals of Statistics*, 49(6):3299–3323.

Lalande, F. and Doya, K. (2023). Numerical data imputation for multimodal data sets: A probabilistic nearest-neighbor kernel density approach. *Transactions on Machine Learning Research*.

Lian, H. (2011). Convergence of functional k-nearest neighbor regression estimate with functional responses. *Electronic Journal of Statistics*, 5:31–40.

Madrid Padilla, O. H., Sharpnack, J., Chen, Y., and Witten, D. M. (2020). Adaptive nonparametric regression with the k-nearest neighbour fused Lasso. *Biometrika*, 107(2):293–310.

Matabuena, M., Vidal, J. C., Padilla, O. H. M., and Onnela, J.-P. (2024). kNN algorithm for conditional mean and variance estimation with automated uncertainty quantification and variable selection. *arXiv preprint arXiv:2402.01635*.

Runge, J. (2018). Conditional independence testing based on a nearest-neighbor estimator of conditional mutual information. In *International Conference on Artificial Intelligence and Statistics*, pages 938–947. PMLR.

Stone, C. J. (1977). Consistent nonparametric regression. *The Annals of Statistics*, 5(4):595–620.

Stone, M. (1974). Cross-validatory choice and assessment of statistical predictions. *Journal of the Royal Statistical Society: Series B (Methodological)*, 36(2):111–133.

Troyanskaya, O., Cantor, M., Sherlock, G., Brown, P., Hastie, T., Tibshirani, R., Botstein, D., and Altman, R. B. (2001). Missing value estimation methods for dna microarrays. *Bioinformatics*, 17(6):520–525.

