# OpenReview forum: "Fast Computation of Leave-One-Out Cross-Validation for $k$-NN Regression"
_TMLR — Accepted by TMLR_

### Review · Reviewer_2heP · 2024-06-04

**Summary Of Contributions:**

The study presents a quick way to calculate leave-one-out cross-validation (LOOCV) in k-nearest neighbors (k-NN) regression. By adding a tie-breaking rule for nearest neighbors, the LOOCV error estimation in k-NN regression matches that of (k + 1)-NN regression, adjusted by a scaling factor of $(k + 1)^2/k^2$. This method allows for efficient LOOCV score computation by training (k + 1)-NN regression once, eliminating the need for repetitive k-NN training-validation loops. Experimental findings validate the effectiveness of this streamlined approach.

**Audience:**

Yes

**Claims And Evidence:**

Yes

**Requested Changes:**

1. Additional elaboration is needed regarding the tie-breaking condition. It would be beneficial to explore whether this condition can be relaxed without compromising the algorithm's effectiveness. This inquiry could shed light on the robustness and flexibility of the approach.


2. The simulation section should explore the implications of not meeting the tie-breaking condition to enhance the versatility of the proposed technique. Furthermore, including additional numerical experiments on real datasets would strengthen the demonstration of the method's efficacy.

**Strengths And Weaknesses:**

Strengths:

1. The paper proposes a rapid computational approach for LOOCV in k-nearest neighbors (k-NN) regression, addressing a practical concern in the field.

2. The incorporation of a tie-breaking condition for nearest neighbors allows for a clear correlation between LOOCV estimate and mean square error in (k + 1)-NN regression, providing a novel insight.

Weaknesses:

1. The paper does not discuss potential limitations or drawbacks of the proposed method.

2. It is unclear whether the tie-breaking condition for nearest neighbors may introduce biases or affect the accuracy of the results.

3. The experimental results, while confirming the efficacy of the method, may not provide a comprehensive evaluation across various datasets or scenarios.

---

> ### Author Response · Authors · 2024-10-21
> **Thank you for your review**
>
> Dear Reviewer,
>
> Thank you for your comments, which helped me revise the paper.
>
> The following is a summary of the revision addressing your comments:
>
> 1. A separate section for discussing the tie-breaking condition is created (Section 5). It studies situations where the tie-breaking condition is not satisfied. The results suggests that the fast method overestimates the LOOCV scores for small $k$, where the true LOOCV scores are large, and thus the best $k$ minimising the LOOCV score is not affected significantly.
>
> 2. Additional experiments are included in Sections 4 and 5 using another real dataset. As in the previous experiments, they show the validity of the fast method for exactly computing the LOOCV scores when the tie-breaking condition is met.
>
> Thank you again, and have a good day.
>
> Best wishes,
>
> Authors

---

### Review · Reviewer_fQbo · 2024-06-13

**Summary Of Contributions:**

This paper proposes a fast way to compute the LOOCV for k-NN.
Under an assumption for tie-breaking, it is shown that there is
simple relationship between the LOOCV score for k-NN and (k+1)-NN.

**Audience:**

Yes

**Claims And Evidence:**

No

**Requested Changes:**

I would suggest that more experiments be performed.

after more validation is provided, the paper can be published.

**Strengths And Weaknesses:**

+ The paper is clearly presented.
 + The results appear new.


- Currently there are only one experiment.

---

> ### Author Response · Authors · 2024-10-21
> **Thank you for your review**
>
> Dear Reviewer,
>
> Thank you so much for your review. Following your comments, the paper has been improved by adding more experiments and validation. Please see the revised version and the summary of the revision above.
>
> Thank you again, and have a good day.
>
> Best wishes,
>
> Authors

---

### Review · Reviewer_TJJm · 2024-10-17

**Summary Of Contributions:**

The paper proposes a computationally efficient way to compute LOOCV for a $k$-NN regression model by establishing the relation between LOOCV of a $k$-NN regression and the training error of a $k+1$-NN model, under a mild tie-breaking condition.
Experiments are performed to validate the relation and the computational benefit.

**Audience:**

Yes

**Claims And Evidence:**

Yes

**Requested Changes:**

N/A

**Strengths And Weaknesses:**

## Strengths
To my best understanding, this finding of the relation between LOOCV of a $k$-NN regression and the training error of a $k+1$-NN model is novel.
The finding is easy to apply and very useful for practitioners to use $k$-NN by selecting $k$ via LOOCV.

## Weaknesses/Questions
The $k+1$-NN computation is essentially computing the $k+1$ nearest neighbors for each data point.
So one might argue, instead of the need of this relation, above is the efficient way to perform LOOCV.
Given this, can we remove the tie-breaking condition?
- If we have the $k+1$ nearest neighbors for each data point, we can actually compute (4) exactly, as we can always selectively removing the data with index $l$ from the neighbor set, which could be done in O(1) if the neighbors are sorted and indexed (if $l$ is in the index, remove it, otherwise remove the farthest neighbor).
With above being said, such algorithm is still based on the main finding of the paper.

---

> ### Author Response · Authors · 2024-10-21
> **Thank you for your review**
>
> Dear Reviewer,
>
> Thank you very much for your comments. As summarised above, the paper has been revised with additional experiments and discussions. Please have a look.
>
> > If we have the $k+1$ nearest neighbors for each data point, we can actually compute (4) exactly, as we can always selectively removing the data with index from the neighbor set, which could be done in O(1) if the neighbors are sorted and indexed ... With above being said, such algorithm is still based on the main finding of the paper.
>
> Yes, if we understand your comment correctly, that would be what the proposed method implicitly does. Thank you for your comment.
>
> > Given this, can we remove the tie-breaking condition?
>
> Thank you for this question.
> Relaxing the tie-breaking condition would still be difficult because the uniqueness of $k$-NN regression and the resulting LOOCV score themselves depend on this condition. If the tie-breaking condition is not met, the $k$-th nearest neighbour may not be unique, so the resulting $k$-NN regression changes depending on how tie-breaking is done (usually done randomly).
>
> Thank you again, and have a good day,
>
> Best wishes,
>
> Authors

---

### Author Response · Authors · 2024-10-18
**Revised Version Updated**

Dear Action Editor and Reviewers,

Thank you so much for your time and comments on this paper.

The paper has been revised based on your comments. The revised parts are highlighted in blue.

The main changes are as follows:

1.  A previous discussion on the tie-breaking condition in Section 4 is improved and separated as a new section (Section 5).
The violation of the tie-breaking condition mainly affects smaller values of k, for which the LOOCV scores are larger.
Therefore, it does not affect significantly the **best** $k$ that **minimizes** the LOOCV score.

2. Additional experiments are added to Sections 4 and 5 using another real dataset. The fast computation method leads to the same LOOCV scores as the brute-force approach, while the computation is much faster, supporting the findings of previous experiments.

We will reply to individual comments soon.
Thank you again, and have a good weekend.

Best wishes,
Authors

---

### Decision · Action_Editor_w1RK · 2024-12-03

**Recommendation:** Accept as is

**Comment:**

I was skeptical of the novelty when I first saw it, but could not find it in the literature, and this was backed up by the reviewers.  The idea has clear merit, thus I feel it should be accepted.

**Audience:**

The topic is clearly on topic for TMLR's audience.

**Claims And Evidence:**

The paper makes a simple, but appears to be novel, insight about leave-one-out cross-validation for k-NN regression.  Once stated, it seems obvious in hindsight -- which is a big positive in my view.  The reviewers agreed, and found the paper well done.